# Histopathological Growth Patterns Determine the Outcomes of Colorectal Cancer Liver Metastasis Following Liver Resection

**DOI:** 10.3390/cancers16183148

**Published:** 2024-09-13

**Authors:** Lucyna Krzywoń, Anthoula Lazaris, Stephanie K. Petrillo, Oran Zlotnik, Zu-Hua Gao, Peter Metrakos

**Affiliations:** 1Cancer Research Program, Research Institute of McGill University Health Center Glen Site, McGill University Health Center, Royal Victoria Hospital-Glen Site, 1001 Decarie Blvd Room E02.6218, Montreal, QC H4A 3J1, Canada; lkrzywon@llu.edu (L.K.); anthoula.lazaris@mail.mcgill.ca (A.L.);; 2Department of Experimental Surgery, McGill University, 1650 Cedar Ave., Room A7.117, Montreal, QC H4A 3J1, Canada; 3Department of Pathology and Labaratory Medicine, University of British Columbia, Rm. G227-2211 Wesbrook Mall, Vancouver, BC V6T 2B5, Canada; 4McGill University Health Center, Royal Victoria Hospital-Glen Site, 1001 Decarie Blvd, Montreal, QC H4A 3J1, Canada

**Keywords:** liver metastasis, colorectal cancer, histopathologic growth patterns, liver resection

## Abstract

**Simple Summary:**

The survival of patients with colorectal cancer liver metastasis can be prognosticated based on the presence or absence of desmoplastic histopathologic growth patterns. Patient survival is improved if the liver tumor has at least a 50% desmoplastic histopathologic growth pattern. Determing the patients tumor histopathological features is essential to prognosis outcomes and treatment management.

**Abstract:**

Introduction: Colorectal cancer liver metastasis (CRCLM) remains a lethal diagnosis, with an overall 5-year survival rate of 5–10%. Two distinct histopathological growth patterns (HGPs) of CRCLM are known to have significantly differing rates of patient survival and response to treatment. We set out to review the results of 275 patients who underwent liver resection for CRCLM at the McGill University Health Center (MUHC) and analyze their clinical outcome, mutational burden, and pattern of cancer progression in light of their HGPs, and to consider their potential effect on surgical decision making. Methods: We performed a retrospective multivariate analysis on clinical data from patients with CRCLM (*n* = 275) who underwent liver resection at the McGill University Health Center (MUHC). All tumors were scored using international consensus guidelines by pathologists trained in HGP scoring. Results: A total of 109 patients (42.2%) were classified as desmoplastic and angiogenic, whereas 149 patients (57.7%) were non-desmoplastic and vessel co-opting. The 5-year survival rates for angiogenic patients compared with vessel co-opting patients were 47.1% and 13%, respectively (*p* < 0.0001). Multivariate analysis showed patients with vessel co-opting CRCLM had a higher incidence of extrahepatic metastatic disease (*p* = 0.0215) compared with angiogenic CRCLM. Additionally, KRAS mutation status was a marker of increased likelihood of disease recurrence (*p* = 0.0434), as was increased number of liver tumors (*p* = 0.0071) and multiple sites of extrahepatic metastatic disease (*p* < 0.0001). Conclusions: Multivariate analysis identified key clinical prognostic and molecular features correlating with the two HGPs. Determining liver tumor HGPs is essential for patient prognostication and treatment optimization.

## 1. Introduction

Approximately 600,000 people die annually of colorectal cancer worldwide, and two-thirds of these deaths are related to liver metastases [1]. The liver is one of the most common sites of colorectal cancer metastases. Approximately 50% of patients will have liver metastases during the course of their disease [2]. The prognosis and treatment of patients with colorectal liver metastasis (CRCLM) has improved dramatically during the past decades [3,4]. This has occurred due to simultaneous advancements in the surgical treatment of CRCLM patients, improvement in modern chemotherapy protocols that downsize tumors and allow resection, and the development of modern antiangiogenic and immunologic drugs.

CRCLM can be characterized by three types of histopathological growth pattern (HGP), defined by the spatial interaction of tumor cells with the surrounding parenchyma at the tumor interface. The desmoplastic HGP (dHGP) is characterized by a capsule of desmoplastic stroma at the tumor interface, separating it from the liver parenchyma. The replacement HGP (rHGP) is characterized by tumor cells infiltrating the hepatic plates of the adjacent parenchyma. In the pushing HGP (pHGP), tumor cells compress the surrounding liver parenchyma but do not infiltrate it [5]. While liver metastases with dHGP depend mostly on angiogenesis to become vascularized, liver metastases with rHGP mostly depend on vessel co-option to supply their nutrients and oxygen. Consequently, liver metastases with a replacement HGP are resistant to antiangiogenic drugs [6,7,8] and have reduced overall survival [9,10]

Multiple studies have demonstrated that liver tumor HGPs are significantly correlated with patient prognosis [6,7,8,11,12,13,14,15,16,17,18,19,20]. While dHGP has consistently been shown to provide a survival benefit compared with non-dHGP, assessment of patient survival with predominantly rHGP vs. pHGP has had mixed results [13,18,19,20,21]. Patients with predominantly dHGP liver tumors are associated with angiogenesis, have more limited liver disease, salvageable recurrences [7], and improved overall survival [8], while patients with non-dHGP liver tumors, which are associated with vessel co-option, do not respond to antiangiogenic drugs [6,22,23,24,25,26], have higher rates of R1 resection, and have reduced overall survival [9,10,27,28,29,30,31,32,33,34].

In this article we will compare the clinical characteristics and survival outcomes of CRCLM patients with dHGP and non-dHGP liver tumors who underwent liver resection at a high-volume hepatobiliary center. We aim to provide new insights regarding the role of liver tumor HGPs in predicting disease free interval, overall survival, and optimal treatment strategy in colorectal cancer patients. An overview of the clinical differences between desmoplastic and non-desmoplastic histopathologic growth patterns are summarized in Table 1 and Appendix A.

### 1.1. Histopathologic Growth Patterns and Primary Tumor Characteristics

The primary CRC tumor may predict the HGP of future CRCLMs. Rajageneshan et al. compared primary CRC tumors with invasive margins characterized as “pushing” as opposed to “infiltrative” with liver tumor in the same patients. They found that patients with “pushing” primary tumors more often developed liver metastases with capsules, while patients with “infiltrative” type primary tumors were more likely to develop non-encapsulated liver tumor. DFS was improved in the “pushing” type but there was no significant difference in OS between the two groups [35].

RHGP liver tumors are associated with high tumor budding scores and infiltrating growth patterns in the primary CRC. Wu et al. assessed features of primary CRC tumors including histology and genetic mutations to see whether it was possible to predict the HGP of CRCLM. Wu also characterized primary tumors in patients with dHGP liver tumors to be associated with expanding growth patterns with low tumor budding scores and a Crohn’s disease-like response. Infiltrative growth patterns in primary CRC and thus rHGP liver metastases were associated with worse OS (*p* = 0.0337) [36].

### 1.2. Histopathologic Growth Patterns, Immune Scores, and Immunotherapy

Recently, work has gone into the development of Immunoscore for the prognostication of different cancers, including CRC. Immunoscore is a standardized scoring system based on lymphocyte populations and densities, such as CD3+ and CD8+, measured at the tumor center and the invasive margin. Scores range from 0 to 4, with higher values associated with longer survival [23]. Immunoscore has been validated as a reliable prognostic predictor for patients with colon cancer stages I-III [37,38]. Studies have found the score to also be reliable following surgical resection of CRCLM with patients who have a high Immunoscore having prolonged RFS and OS [23]. Liang et al. developed a scoring system combining liver tumor HGP, Immunoscore, and CRS after they found the densities of CD+3 and CD+8 immune cells were higher in dHGP compared with non-dHGPs liver tumors [23].

CRCLM tumors have been evaluated for individual tumor immune phenotypes and have demonstrated vessel co-opting tumors to be associated with lower immune reactions compared with desmoplastic tumors. Stremitzer et al. evaluated immune phenotypes of CRCLM patients by analyzing the results of patients who were treated with perioperative bevacizumab-based chemotherapy and found that desmoplastic liver tumors were associated with an “inflamed” immune phenotype, shown by the presence of CD8-positive immune cells at the tumor interface and the tumor itself. In contrast, replacement HGP were found to have no CD8-positive T-cell infiltration in the tumor and deemed to have a “non-inflamed” or “desert” immune phenotype. The desmoplastic HGP was associated with better radiologic response compared with the replacement HGP subgroup, along with having better histological response rates and longer RFS and OS [23].

Compared with dHGP CRCLM, vessel co-oping rHGP had less lymphocytic infiltration and less expression of immune-related genes such as CD8A/CD8B, GZMA/GZMB, and PRF1 [23]. Therefore, it is possible that non-dHGP liver tumors will have a decreased response to immunotherapy compared with dHGP, resulting in a worse prognosis.

### 1.3. Vessel Co-Option and Genetics

Identifying mutations in liver tumors of patients with CRCLM can guide management by predicting responses to targeted therapy. Thus far, there is no documented association between liver HGP and actionable mutations in the liver tumor. However, a number of genes identified via RNA-sequencing have been associated with vessel co-option and rHGP. Lazaris et al. identified a number of differentially expressed genes using RNA-sequencing data taken from rHGP lesions that were associated with cell migration, cell motility, proteolysis, and wound healing. These included CXCL9, LOXL4, PTHLH, TMEM156, and TNFRSF12A. When compared with dHGP, LOXL4 was significantly upregulated in vessel co-opting lesions (*p* = 0.0015) [39]. Currently, LOXL4 has not been studied in CRCLM, but it is known to function as a catalyst for the crosslinking of collagen and elastin in extracellular matrix remodeling, which may contribute to cancer metastasis [37].

The mutational burden of primary CRC tumors may predict the HGP of future liver metastases. Wu et al. [23] found primary CRC tumors with infiltrating growth patterns were associated with rHGP liver tumors. Both infiltrating-type primary CRC lesions and rHPG CRCLM were associated with mutations in APC (⅗) and TP53 (⅗), KRAS, FAT4, DNH5, SMAD, ERBB2, ERBB3, LRP1, and SDK1 (⅕). On the other hand, dHGP was associated with primary CRC tumors with expanding growth patterns and mutations in APC (⅘); TP53 (⅗); KRAS, PIK3CA, and FAT4 (⅖); and BRCA-1, BRCA2, BRAF, and DNAH5 (⅕). They also found PIK3CA, a mutation in the PIK3 pathway that improves cell proliferation, leading to carcinogenesis and being associated with angiogenesis. The PIK3CA mutation was present in 40% of CRCLM with dHGP, which may support desmoplastic tumor growth via angiogenesis [23].

An overview of the clinical differences between desmoplastic and non-desmoplastic histopathologic growth patterns are summarized in Table 1 and Appendix A.

## 2. Methods

### 2.1. Patient Selection

We performed a retrospective data collection on all patients 18 years and older, with CRCLM, who had consented to participate in the McGill University Liver disease biobank research program. The study participants were all living in the province of Quebec at the time of recruitment into the McGill University Liver disease biobank. Of the consented patients, we excluded those who did not undergo surgical resection of their liver metastases, as the histopathologic growth pattern could not be determined via biopsy alone. There were 352 patients with colorectal liver metastases who consented and were included in the liver biobank. Of those, 277 patients, 176 male (63.5%) and 101 female (36.4%), underwent liver resection. The liver resections were completed between January 2009 and December 2020 at McGill University Health Center (MUHC). Sixty-six patients who only underwent liver biopsy were excluded from the analysis, as a biopsy is insufficient for determining tumor HGPs. Of the 277 patients, 258 had their liver tumors scored for HGPs. A total of 109 patients (42.2%) were designated as desmoplastic and 149 patients (57.7%) were designated as non-desmoplastic at their initial liver resection. The HGPs of non-desmoplastic patients were either rHGP, pHGP, or mixed (Figure 1). All patients were followed until death. Loss of follow up was defined as no patient contact for 12 months. Patient data were updated and reviewed through July 2022.

All patients were intended to be followed until death. Loss of follow up was defined as no patient contact for 12 months. Patient data was updated and reviewed through July 2022.

### 2.2. Determination of Tumor Histopathologic Growth Pattern

Liver tumor HGPs were scored by AL and ZG, who were both involved in outlining the international consensus guidelines for scoring the HGPs of liver metastasis [12]. HGPs were scored either immediately following surgical resection, or retrospectively scored by reviewing all relevant H&E slides of each patient’s entire resected liver tumor. All resected liver tumors from each patient were evaluated for HGP using the international consensus guidelines [12]. There was no minimum section required for pathologic assessment and the entire tumor interface was evaluated. Tumors with greater than 50% of a specific growth pattern, i.e., dHGP, rHGP, or pHGP, were designated predominately HGP. If a patient had multiple liver tumors with different dominant growth patterns, or if a patient had liver tumors with >2 HGPs, the patient would then be designated as “mixed” and grouped with the non-desmoplastic patients. There was no minimum section required for pathologic assessment.

### 2.3. Next Generation Sequencing

Genomic sequencing was performed either as a send-out test for specific PCR testing of KRAS, NRAS with or without BRAF, or using the Illumina AmpliSeq focus panel for solid tumors utilized by the McGill University Pathology Department. NGS and/or PCR testing was performed on both the primary colorectal tumor and at least one metastatic liver tumor. Tissue sections were assessed on tumor regions with more than 40% viability. All patients in this study were MSI stable.

### 2.4. Statistical Analysis

Categorical data were reported both as absolute numbers and corresponding percentages and compared using the Chi-squared test. Non-parametric continuous data were reported as median values with corresponding standard deviation values (SD). Kaplan–Meier analysis with log-rank tests were used to estimate survival curves. Univariate and multivariate Cox proportional hazard regression survival analyses were performed and reported as hazard ratios (HRs) with corresponding 95% confidence intervals (CIs) and *p*-values. An alpha value of 0.05 was used to indicate statistical significance. The statistical calculations were performed using SAS version 9.4 statistical software.

## 3. Results

At McGill University Health center, there were 352 patients with colorectal liver metastasis who consented for participation in the liver biobank research program. Of these patients, 277 underwent liver resection between January 2009 and December 2020 at McGill University Health Center. All 277 patients had their liver tumors evaluated for histopathologic growth patterns, but 19 patients were unable to have their liver tumors scored due to completely mucinous tumors, or no evidence of residual malignancy. Of the remaining 258 patients, 109 had desmoplastic tumors (42.2%) and 149 patients had non-desmoplastic tumors (57.7%). There was no significant difference between desmoplastic and non-desmoplastic patients when looking at gender, age of diagnosis, BMI, or incidence of synchronous disease. Unfortunately, the race and ethnicity of patients were not at the time being consistently documented in the electronic medical record of patients at the MUHC and therefore were not included in the statistical analysis. There was no significant difference between the desmoplastic and non-desmoplastic groups when comparing the size and location of the primary tumor; however, there was a significant difference in the size of the liver metastasis. Non-desmoplastic tumors were slightly larger than desmoplastic tumors when measuring the tumor’s greatest dimension. The mean greatest dimension of desmoplastic liver tumors was 3.34 cm (SD 2.35) and the mean greatest dimension of non-desmoplastic tumors was 3.86 cm (SD 2.78) (*p* = 0.047). There was a trend towards a higher number of tumors in patients with non-desmoplastic HGP (*p* = 0.0581). The mean number of tumors in patients with desmoplastic HGPs was 3.07 (SD 2.35) and the mean number of liver tumors in patients with non-desmoplastic HGPs was 3.4 (SD 2.03) (Table 2).

### 3.1. Systemic Therapy

Overall, 24.3% (N = 25) of dHGP patients were chemo-naive and 19.9% (N = 27) of non-dHGP patients were chemo-naive. For the remaining dHGP patients, 35.9% (N = 28) received neoadjuvant bevacizumab compared with 41.3% (n = 45) of the non-dHGP cohort. There was no significant difference in the number of patients who were chemo-naive or who received neoadjuvant chemotherapy in the dHGP and non-dHGP groups. There was also no difference in the number of patients who received neoadjuvant bevacizumab (Table 2). The most common systemic chemotherapy regimens were FOLFOX (N = 170), followed by FOLFIRI (N = 18) and XELODA (N = 9).

### 3.2. Mutation Analysis

We compared the number of mutations found in the liver tumor and found in the primary tumor of patients with dHGP and non-dHGP. There was no significant difference in the incidence of KRAS, NRAS, or BRAF mutations identified in the liver tumors or in the primary tumors in the desmoplastic group compared with the non-desmoplastic group (Appendix A). There was no significant difference in the incidence of overall number of mutations found in the liver tumors or the primary tumors when comparing the desmoplastic and non-desmoplastic groups (Appendix A).

We also evaluated the mutational burden of the primary tumors. In our analysis, right-sided colon tumors had a statistically higher incidence of liver tumors with KRAS mutations (*p* = 0.0241) when compared against left-sided tumors and rectal tumors (Appendix A). There was a significant difference in PIK3CA mutational burden based on the location of the primary tumor (*p* = 0.0139), with most PIK3CA mutations located in right-sided primary tumors. Seven out of the ten patients with PIK3CA mutations had right-sided colon tumors (Appendix A). When looking at the overall mutational burden of the liver tumors, there was a significant difference in the location of the associated primary tumor (*p* = 0.0042); however, there was no significant difference in the overall mutational burden of the primary tumor (*p* = 0.4026). Also, there was no significant difference in the location of the primary tumor of desmoplastic vs. non-desmoplastic liver tumors (*p* = 0.8261) (Appendix A).

A subgroup survival analysis was performed for all patients with right-sided primary tumors. Interestingly, when comparing overall survival from the time of liver metastasis diagnosis, the median survival of dHGP patients (N = 31) was 60 months, and the median survival of non-dHGP patients (N = 37) was 39 months (*p* = 0.0231) (Appendix A).

### 3.3. Survival Analysis

The median survival for the entire cohort of 258 patients who had their liver tumor HGP scored was 66 months from initial diagnosis of CRC and 49 months from diagnosis of CRCLM. A total of 71.3% of patients presented with synchronous disease. Synchronous disease was defined as the presence of liver metastasis within 1 year of CRC diagnosis.

The median survival for patients with dHGP liver tumors from diagnosis of CRC and CRCLM was 79 and 61 months, respectively, ranging from 11 to 192 months. The median survival for patients with non-dHGP from diagnosis or CRC and CRCLM was 56 and 43 months, respectively, ranging from 3 to 146 months. For patients with dHGP, the 3 and 5 year OS after CRC diagnosis was 73.8% and 45.8%, respectively, and the 3 and 5 year OS after diagnosis of CRCLM was 65.4% and 34.6%, respectively. For patients with non-dHGP, the 3 and 5 year OS after diagnosis of CRC was 62.3% and 39.8%, respectively, and the 3 and 5 year OS after diagnosis of CRCLM was 47.1% and 13%, respectively.

Kaplan–Meier curves found the dHGP patients to have a significantly greater OS compared with non-dHGP patients when evaluating survival from diagnosis of CRC (*p* = 0.0182) and CRCLM (*p* < 0.0031) (Figure 2A and Figure 2B, respectively). Patients with synchronous disease had worse overall survival regardless of HGP (*p* < 0.0359) (Figure 2C).

There was no significant difference between the proportion of dHGP and non-dHGP groups who received neoadjuvant chemotherapy (*p* = 0.4123) or neoadjuvant chemotherapy + bevacizumab (*p* = 0.4565), as seen in Table 2. There was no significant difference between the survival curves evaluating chemotherapy regimens (chemo-naive vs. neoadjuvant chemotherapy vs. neoadjuvant chemotherapy + bevacizumab) (Figure 3A, *p* = 0.1074). There was no significant difference in OS for dHGP patients and non-dHGP patients who were chemo-naive or who received neoadjuvant chemotherapy (*p* = 0.1445) or neoadjuvant chemotherapy + bevacizumab (*p* = 0.9335) (Figure 3B and Figure 3C, respectively). There was a trend towards significance when comparing the OS of patients who received neoadjuvant chemotherapy + Bevacizumab (*p* = 0.0767) with dHGP patients exhibiting improved overall survival compared with non-dHGP patients (Figure 3D). We excluded patients who required two-staged hepatectomy.

Interestingly, when we evaluated the overall survival of patients based on their percent desmoplastic HGP, we found a statistically significant difference between patients with 90–100% desmoplastic HGP, patients with 50–89% desmoplastic HGP, and patients with under 50% HGP (or non-desmoplastic HGP, (*p* = 0079) (Figure 2D).

### 3.4. Extrahepatic and Recurrent Disease

There was a significantly higher incidence of extrahepatic metastatic disease in non-desmoplastic patients compared with desmoplastic (*p* = 0.0009) and a statistically higher incidence of pulmonary metastatic disease (*p* = 0.0006) (Table 1). There was a trend towards a higher incidence of multiple sites of metastatic disease in non-desmoplastic patients (*p* = 0.1575) with 42.1% of non-desmoplastic patients having multiple sites of recurrent disease compared with 33.3% of desmoplastic patients. When evaluating liver-specific recurrence rates, 44% of desmoplastic patients never experienced liver recurrence, compared with only 30.6% of non-desmoplastic patients. Also, 22.4% of non-desmoplastic patients were unable to have their liver tumors fully resected, compared with only 13% of desmoplastic patients (Table 2) (*p* = 0.0375)

Factors associated with recurrent metastatic liver disease and inability to fully resect liver tumors included larger size of liver tumor (4.05 cm vs. 2.94 cm vs. 3.9 cm, *p* = 0.0037), higher number of liver tumors at initial diagnosis (3.52 vs. 2.23 vs. 4.74, *p* < 0.0001), synchronous presentation (*p* = 0.0018) and receiving neoadjuvant chemotherapy prior to colon resection (*p* = 0.0120) or liver resection (*p* = 0.0024), having non-desmoplastic HGP (*p* = 0.0375) development of pulmonary metastasis (*p* < 0.0001), and extrahepatic metastasis (*p* < 0.0001) (Appendix A).

### 3.5. Univariate and Multivariate Analysis

Regarding univariate analysis (UVA) controlling for patient age, gender, and BMI, higher OS was associated with dHGP (*p* = 0.0010), metachronous presentation (*p* < 0.0001), and chemo-naive status prior to undergoing colon resection (*p* < 0.0001). Regarding multivariate analysis (MVA), only metachronous presentation (*p* = 0.002) and chemo-naive status prior to colon resection (*p* = 0.011) were statistically significant for improved OS. Development of extrahepatic metastasis, multiple sites of extrahepatic metastasis, and pulmonary metastasis were all associated with worse OS regarding both UVA and MVA (Table 2).

Cancer recurrence was associated with non-desmoplastic HGP (*p* = 0.0001) and presence of neoadjuvant chemotherapy prior to colon resection or liver resection (*p* = 0.0100 and *p* = 0.024, respectively) regarding both UVA and MVA controlling for patient age, gender, and BMI. Regarding UVA, KRAS status was associated with cancer recurrence (0.0392), but not regarding MVA. Regarding both UVA and MVA, development of extrahepatic metastatic disease, pulmonary metastatic disease, and multiple sites of metastatic disease were all associated with increased risk of cancer recurrence following liver resection (Table 3).

## 4. Discussion

This study aimed to evaluate the prognostic differences between desmoplastic and non-desmoplastic HGP in the setting of CRCLM. We retrospectively analyzed all the HGPs in patients at McGill University Health Center with CRCLM who underwent liver resection.

According to our results, dHGP was independently associated with improved OS and a lower overall incidence of cancer recurrence. Our study showed a dramatic improvement in both 3 and 5 year survival for patients with predominantly dHGP after diagnosis of CRCLM, as demonstrated by reporting a median 3 year OS of 65.4% for dHGP compared with 47.1% for non-dHGP and a median 5 year OS of 34.6% for dHGP patients compared with 13% for non-dHGP patients.

Galjart et al. published the largest single center study to date investigating the prognosis of CRCLM patients based on their liver tumor HGP [8]. They also reported improved survival in patients with angiogenic dHGP liver tumors, (78% 5-year survival). However, they used a cut-off value of 100% to categorize dHGPs. While many other groups have previously reported improved survival for patients with high percentages of dHGP, we were able to demonstrate improved outcomes for patients with >50% dHGP according to the international consensus guidelines for scoring CRCLM, and as previously documented by Frentzas et al. [6]. This is significant because the majority of patients develop multiple liver tumors with mixed phenotypes. We have demonstrated that patients with CRCLM demonstrating at least 50% of the interface demonstrating a desmoplastic ring can have a similar improved prognosis as patients with a “pure” desmoplastic phenotype. We were also able to show that there was a significant difference in survival between patients with different dHGP cut-off points (*p* = 0.0079, Figure 2D). Patients had improved OS with both 50–89% dHGP and >90% dHGP compared with non-desmoplastic HGPs. Including patients with 50–89% dHGP allowed us to double our dHGP cohort while still improving overall patient survival.

When we performed univariate and multivariate analysis comparing patients based on their systemic chemotherapy regimens, patients who received neoadjuvant chemotherapy with or without bevacizumab prior to undergoing liver resection had a worse overall survival (*p* = 0.1074) (Figure 3). Half of the chemo-naive cohort were dHGP (n = 20) compared with non-dHGP (n = 20), and 42% (n = 21) had a single liver lesion. We hypothesize that the chemo-naive patients had a lower burden of disease compared with patients who received either neoadjuvant chemotherapy + bevacizumab or neoadjuvant chemotherapy alone. Therefore, clinicians may have opted to perform upfront surgical resection. Interestingly, there was a near significance in improved overall survival comparing patients who received neoadjuvant chemotherapy with bevacizumab (*p* = 0.0767, Figure 3). This cohort mirrored Frentzas et al.’s cohort, as we selected patients who had complete R0 resections of the liver metastasis.

Unfortunately, the majority of patients with CRCLM develop recurrent disease. This was especially true in our cohort, which exhibited significantly higher rates of synchronous disease compared with other publications. The rate of synchronous disease was 71% in our cohort (Table 2). We were able to demonstrate a different pattern in disease recurrence among dHGP patients and non-dHGP patients. In UVA and MVA, non-dHGP was associated with cancer recurrence (*p* < 0.0001, HR 1.92 and *p* = 0.0001, HR1.83, respectively, Table 2). Patients with non-dHGP were more likely to develop extrahepatic metastasis (*p* = 0.0006, HR 1.82 on UVA; *p* = 0.0007, HR 1.88 on MVA) and were more likely to develop pulmonary metastasis (*p* < 0.0001, HR 2.02 and 2.01 on UVA and MVA, respectively, Table 3) and develop multiple sites of extrahepatic metastasis (*p* < 0.0001, HR 2.09 and 2.16 on UVA and MVA, respectively, Table 3). There was a higher percentage of non-dHGP patients who developed disease at multiple extrahepatic locations (33.3% vs. 42.1%) (Appendix A), but it was not significant (*p* = 0.1575). These findings are complemented by Neirop et al., who previously reported that patients with dHGP were more likely to develop liver recurrence and were more likely to have recurrent liver disease amenable to re-operation [10]. We think this is a large part of the reason for patient survival improving for patients with desmoplastic tumors. Since desmoplastic tumors are more likely to recur in the liver, patients are more likely to have repeat liver operations or procedures—such as RFA—to control the recurrence. This is in contrast with patients with non-desmoplastic tumors who are more likely to have extra-hepatic disease recurrence, which is non-resectable. These findings suggest that overall survival in patients with CRCLM is driven more by rHGP and vessel co-option than dHGP angiogenesis.

Limitations of this study include the retrospective nature of data collection and the lack of validation at an external site. We also had not been collecting race/ethnicity data on our liver disease biobank subjects until the end of 2021. We are in the process of adding race and ethnicity documentation for each subject. Additionally, molecular testing of liver tumors for all CRCLM patients was recently implemented at McGill University, therefore only 40% (n = 98) of our cohort was able to be analyzed for mutations. With uniform testing for multiple mutations, we hope to be able to determine an optimal targeted therapy approach for patients with different HGPs. So far, we have identified a higher incidence of mutations in non-dHGP patients and inability to fully resect the liver tumors (*p* = 0.0375, Table 4) and a higher incidence of KRAS mutations in patients with right-sided colon tumors (*p* = 0.0241 and *p* = 0.0197, Appendix A) and an association of PIK3CA mutations with right-sided primary tumors (*p* = 0.0139, Appendix A). It has been documented in numerous studies that right-sided colon tumors are associated with more advanced tumor stages, increased tumor sizes, and poor differentiation. All these findings are likely due to right-sided tumors becoming symptomatic later than left-sided tumors due to the wider lumen of the left colon and possibly due to the higher difficulty in identification of the right-sided tumors as they are far from the anal verge and unable to be identified on physical exam or sigmoidoscopy. Taking these factors together, right-sided colon tumors have been found in other studies to convey a worse overall survival compared with left-sided colon tumors. To date, the mechanistic link between KRAS mutation and tumor sidedness also remains to be assessed. These findings will need to be validated in future studies with a larger patient cohort. We do not believe these limitations affected the overall conclusions of this study.

## 5. Future Directions

Our cohort population presented in this study is biased towards patients who were able to undergo at least partial resection of their metastatic liver tumors. However, in general, only 10–20% of patients with CRCLM are resectable [9]. There has been consistent evidence associating worse survival for patients with non-dHGP CRCLM. Currently, it is unknown why a patient develops dHGP vs. non-dHGP liver tumors. Thus, it is essential to accurately identify the biomarkers and characteristics of the tumor microenvironment of early developing liver tumors to guide treatment and identify optimal downstaging treatments.

Early HGP identification can guide treatment and identify optimal downstaging therapy to allow for liver resection. Development of imaging and liquid biopsy tools are essential to accurately identify the liver tumor HGP early in the disease course. Biomarkers identified by RNA sequencing in liquid biopsies could predict CRCLM HGP and guide systemic and immunologic therapies for downstaging. Liver tumor tissue biopsies might also be able to predict HGP in the future despite not providing the full tumor interface. A liver biopsy can provide enough tissue to perform next generation sequencing and provide information on the tumor immune population, which has been shown to differ between dHGP liver tumors and non-dHGP liver tumors, as previously discussed [9]. Radiologic features seen on MRIs of liver tumors in CRCLM have been able to predict the presence of KRAS mutations [38], and dHGPs appear to have a fibrous band encircling tumors visible on contrast-enhanced CT scans [40].

Neoadjuvant systemic therapy regimens could be tailored for patients based, in part, on their HGP. Patients with rHGP might benefit from more aggressive perioperative treatment without the addition of bevacizumab. Surgical planning could be influenced by pre-operatively knowing the tumor HGP. Patients with known rHGP might benefit from a surgeon intentionally performing anatomical liver resections or taking a wider margin to decrease the risk of performing an R1 resection. This might also lead to more two-staged hepatectomies for patients with bilateral disease and rHGP tumors due to the need to develop an adequate future liver remnant. On the other hand, patients with known dHGP might tolerate parenchymal sparing resections.

Additionally, liver transplantation for highly selected patients with unresectable liver-limited metastatic CRC has shown promising survival results [41,42,43]. The 5 year OS for patients with CRCLM following liver transplantation was 75% compared with a 76% 5 year OS for patients with HCC [41,42]. Patients with dHGP, who are less likely to have extra-hepatic metastatic disease and more likely to have liver-limited recurrent disease, would likely benefit from liver transplantation. Since dHGP liver tumors were found to have a higher immune response, it remains unclear what effect immunosuppression would have on cancer recurrence. Of note, the SECA investigators did not find a difference in the growth rate of lung nodules in patients with CRCLM treated with liver transplantation and immunosuppression compared with CRCLM patients who did not undergo liver transplantation [44]. Also, rapamycin demonstrated antiangiogenic qualities to inhibit tumor growth following hepatectomy for CRCLM [41], and sirolimus-based immunosuppression has demonstrated improved OS, specifically for patients with HCC following liver transplantation [45,46].

## 6. Conclusions

In conclusion, this study validates the improved prognostic value of the desmoplastic HGP in patients with CRCLM when categorizing patients according to the current international consensus guidelines for scoring HGPs in CRCLM. We recommend that all patients with CRCLM undergo HGP scoring of all liver tumors at each liver resection. Patients with CRCLM are highly complex and should be discussed at multidisciplinary team discussions where personalized treatment plans can be developed.

## Figures and Tables

**Figure 1 cancers-16-03148-f001:**
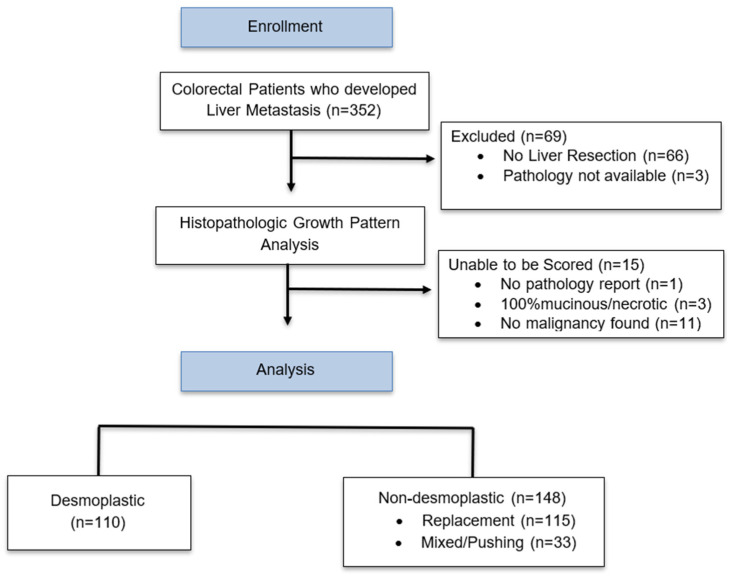
Flowchart of patient enrollment and stratification.

**Figure 2 cancers-16-03148-f002:**
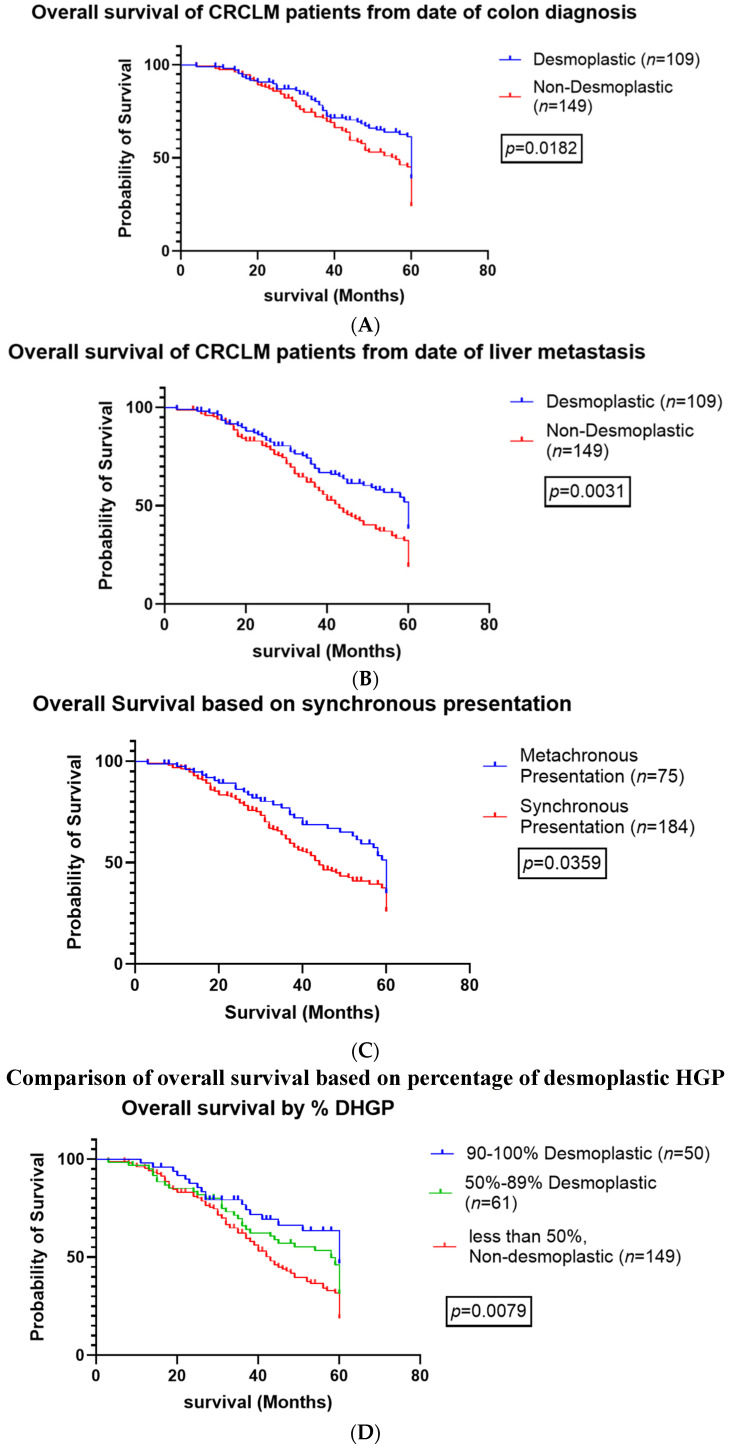
Overall survival based on histopathologic growth patterns. (**A**). Overall survival of patients based on the date of primary colon cancer diagnosis. (**B**). Overall survival of patients based on the date of liver metastasis diagnosis (**C**). Overall survival based on synchronous presentation of colon cancer with liver metastasis (**D**). Overall survival based on percent of tumor DHGP.

**Figure 3 cancers-16-03148-f003:**
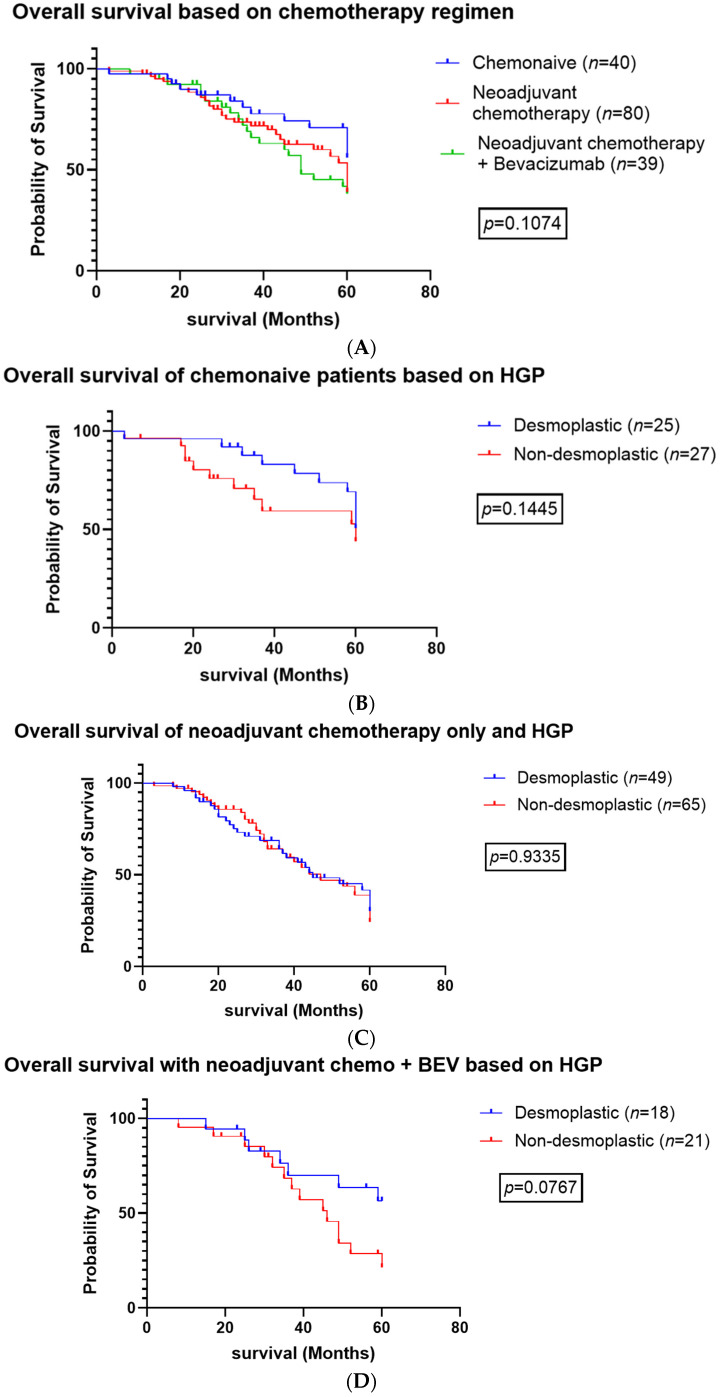
Overall survival comparing systemic chemotherapy regimens. (**A**). Overall survival of patients based on chemotherapy regimens (**B**). Overall survival of chemonaive patients based on histopathologic growth patterns (**C**). Overall survival of patients who received neoadjuvant chemotherapy without Bevacizumab based histopathologic growth patterns (**D**). Overall survival survival of patients who received neoadjuvant chemotherapy with Bevacizumab based histopathologic growth patterns.

**Table 1 cancers-16-03148-t001:** Summary of clinical differences between desmoplastic and non-desmoplastic histopathologic growth patterns.

	Desmoplastic	Non-Desmoplastic
Survival analysis	Increased OS, MS, DFS, PFS	Decreased OS, MS, DFS, PFS
Surgical outcomes	Increased rate of successful re-resection for recurrent disease	Increased risk of R1 resection; increased risk of incomplete resection
Response to systemic chemotherapy	Good response to chemotherapy	Decreased response
Response to targeted therapy	Good response	Poor response to anti-VEGF and anti-EGFR therapy
Disease recurrence	Lower rate of overall disease recurrence; increased rate of hepatic recurrence compared with extrahepatic recurrence	Increased rate of overall recurrence; increased rate of extrahepatic recurrence compared with hepatic recurrence
Immune landscape	Increased lymphocyte infiltration; increased numbers of CD3^+^ and CD8^+^ immune cells	Decreased lymphocyte infiltration; adaptive immune phenotype with neutrophils present
Primary colon tumor	Lower tumor budding score; pushing colon tumor margin	High tumor budding score; infiltrative colon tumor margin

OS—overall survival, MS—median survival, DFS—disease-free survival, PFS—progression-free survival.

**Table 2 cancers-16-03148-t002:** Overall survival using Cox regression.

Characteristics	Univariate			Multivariate		
	Hazard Ratio	95% CI	*p*-Value	Hazard Ratio	95% CI	*p*-Value
**Liver tumor histology**						
Desmoplastic	1.00			1.00		
Non-desmoplastic	1.70	1.23–2.34	**0.0010**	1.54	1.09–2.15	**0.0133**
**Synchronous presentation**						
No	1.00			1.00		
Yes	4.30	2.79–6.65	**<0.0001**	4.13	2.63–6.48	**<0.0001**
**Neoadjuvant chemotherapy before primary resection**						
No	1.00			1.00		
Yes	1.87	1.36–2.55	**<0.0001**	1.80	1.30–2.50	**0.0004**
**Adjuvant chemotherapy after liver resection**						
No	1.00			1.00		
Yes	1.51	0.82–2.83	0.1819	1.54	0.80–2.98	0.1960
**TNM stage of primary tumor**						
1	1.00			1.00		
2	1.67	0.39–7.13	0.4875	1.82	0.42–7.79	0.4208
3	2.96	0.43–12.09	0.1299	2.76	0.67–11.29	0.1588
4	4.98	1.20–20.69	**0.0273**	5.92	1.41–24.77	**0.0149**
**KRAS**						
No	1.00			1.00		
Yes	1.43	0.99–2.09	0.0600	1.56	1.03–2.36	**0.0373**
**Location of primary tumor**						
Bilateral	1.00			1.00		
Right	3.75	0.91–15.39	0.0670	3.32	0.79–14.00	0.1022
Left	2.26	0.55–9.32	0.2592	2.00	0.48–8.34	0.3442
Rectum	2.53	0.62–10.34	0.1950	2.13	0.52–8.78	0.2975
**Volume of primary tumor**						
	1.003	0.995–1.010	0.4924	1.002	0.995–1.010	0.5749
**Greatest dimension of liver tumor**						
	0.968	0.912–1.028	0.4822	0.3673	0.907–1.029	0.2803
**Development of extra-hepatic metastatic disease**						
No	1.00			1.00		
Yes	1.82	1.29–2.57	**0.0006**	1.88	1.3–2.72	**0.0007**
**Multiple of extra-hepatic metastatic site**						
No	1.00			1.00		
Yes	1.96	1.44–2.67	**<0.0001**	2.02	1.46–2.80	**<0.0001**
**Liver metastatic recurrence**						
No	1.00			1.00		
Yes	1.46	1.02–2.10	**0.0403**	1.81	1.13–2.88	**0.0127**
Never fully resected	3.42	2.25–5.22	**<0.0001**	1.32	0.79–2.22	0.2871
**Liver resection with and without BEV**						
Chemonaive	1.00			1.00		
Neoadjuvent	2.26	1.42–3.59	**0.0005**	1.93	0.97–3.82	0.0602
BEV	1.68	1.02–2.76	**0.0415**	2.98	1.47–6.02	**0.0024**
**Number of clinically relevant mutations in liver tumors**						
0	1.00			1.00		
1	1.76	1.10–2.81	**0.0177**	2.06	1.24–3.43	**0.0053**
2	1.20	0.64–2.27	0.5666	1.45	0.73–2.87	0.2912
3	13.42	1.71–105.69	**0.0136**	16.81	2.00–141.45	**0.0094**
**Number of clinically relevant mutations in primary tumor**						
0	1.00			1.00		
1	1.42	0.80–2.52	0.2326	1.74	0.93–3.28	0.0838
2	1.21	0.55–2.64	0.6412	1.62	0.72–3.67	0.2459
3	39.98	3.56–448.71	**0.0028**	63.79	4.62–881.05	**0.0019**

Models are all adjusted for age at first diagnosis, gender, and BMI. Significant values are highlighted in bold.

**Table 3 cancers-16-03148-t003:** Cancer recurrence using cox regression.

Characteristics	Univariate			Multivariate		
	Hazard Ratio	95% CI	*p*-Value	Hazard Ratio	95% CI	*p*-Value
**Number of liver tumors at 1st diagnosis**						
	1.103	1.045–1.163	**0.0003**	1.095	1.035–1.158	**0.0016**
**Greatest dimension of liver tumor**						
	1.034	0.954–1.056	0.8781	0.999	0.947–1.054	0.9685
**Development of extra-hepatic metastatic disease**						
No	1.00			1.00		
Yes	2.03	1.49–2.78	**<0.0001**	2.10	1.50–2.93	**<0.0001**
**Development of pulmonary metastasis**						
No	1.00			1.00		
Yes	2.02	1.50–2.72	**<0.0001**	2.01	1.46–2.93	**<0.0001**
**Liver metastasis recurrence**						
No	1.00			1.00		
Yes	1.37	0.99–1.89	0.0543	1.35	0.96–1.90	0.0866
**Multiple of extra-hepatic metastatic site**						
No	1.00			1.00		
Yes	2.09	1.57–2.77	**<0.0001**	2.16	1.60–2.91	**<0.0001**
**Resection with and without BEV**						
Chemo-naive	1.00			1.00		
Neoadjuvent	1.89	1.27–2.80	**0.0018**	1.61	1.07–2.42	**0.0230**
BEV	1.27	0.83–1.95	0.2748	1.07	0.68–1.69	0.7589

Models are all adjusted for age at first diagnosis, gender, and BMI. Significant values are highlighted in bold.

**Table 4 cancers-16-03148-t004:** Patient characteristics of desmoplastic vs. non-desmoplastic tumors.

	Desmoplastic	Non-Desmoplastic	*p*-Value
	*n* = 110	*n* = 148	
Age at diagnosis, mean (SD)	62.4 ± 10.22	60.57 ± 10.13	0.1250
BMI, mean (SD)	26.80 ± 5.66	26.96 ± 4.63	0.6221
Mean size of primary tumor, cm (SD)	13.79 ± 29.14	8.03 ± 11.76	0.1375
Number of liver tumors at 1st diagnosis, mean (SD)	3.07 ± 2.35	3.41 ± 2.03	0.0592
Volume of primary tumor, mean (SD)	13.79 ± 29.14	7.98 ± 11.69	0.1372
Greatest dimension of liver tumor cm (SD)	3.34 ± 2.35	3.86 ± 2.78	**0.0470**
Number of liver tumors at initial diagnosis	3.07 ± 2.35	3.40 ± 2.03	0.0581
Synchronous presentation	82 (74.5%)	101 (68.2%)	0.2702
**Systemic therapy**			
Chemo-naive	25(24.3%)	27 (19.9%)	0.4123
Neoadjuvant chemotherapy	78 (75.7%)	109 (80.1%)	0.4123
Neoadjuvant chemotherapy + bevacizumab	28 (35.9%)	45 (41.3%)	0.4565
**Metastatic disease**			
Development of extrahepatic metastasis	59 (53.6%)	106 (73.6%)	**0.0009**
Multiple extrahepatic metastatic sites	36 (33.3%)	61 (42.1%)	0.1575
Development of pulmonary metastasis	51 (46.4%)	97 (67.8%)	**0.0006**
**Liver metastatic recurrence**			
Yes	46 (29.5%)	69 (39.2%)	**0.0375**
No	48 (44.4%)	45 (30.6%)	
Never fully resected	14 (13.0%)	33 (22.4%)	

N—patient number, SD—standard deviation. Categorical variables were compared using Chi-squared test. Significant values are highlighted in bold.

## Data Availability

The raw data supporting the conclusions of this article will be made available by the authors on request.

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
