# Peer review of "Histopathological Growth Patterns Determine the Outcomes of Colorectal Cancer Liver Metastasis Following Liver Resection"

_cancers, 2024, doi:10.3390/cancers16183148_

Round 1

Reviewer 1 Report

Comments and Suggestions for Authors

The title accurately reflects the content of the study. The abstract states the main research goals and obtained results. The introduction is well-written although a bit long. The authors should consider transferring some parts to the discussion or leaving them out. Materials and methods are not detailed for the mutation detection part I presume because it was an outsourced service. However, it would be interesting to know how was the tumor heterogeneity addressed because it is known that different parts of tumors harbour different mutations.   The results are well presented.  The MSI status is also an important factor in determining the choice of therapy as well as patient outcomes- was this status known and evaluated? Given that the samples were obtained from the biobank, the mutation analysis was outsourced and as I understand only the histopathology was done for this study in-house by one of the authors, perhaps a more detailed contribution of other authors to the study would be appreciated. In total, it is a well-performed study with new findings valuable for researchers and clinitians. 

Comments on the Quality of English Language

Minor language editing due to grammar and style issues.

Author Response

Comment 1: The authors should consider transferring some parts to the discussion or leaving them out.

Response: We have made changes to the discussion and included some of the sections in the results in the discussion.

Comment 2: Materials and methods are not detailed for the mutation detection part I presume because it was an outsourced service. However, it would be interesting to know how was the tumor heterogeneity addressed because it is known that different parts of tumors harbour different mutations. 

Response: Thank you for your comment. Mutation detection was limited in the earlier patient populations until about 2021, at which time we initiated in house NGS using the Illumina focus panel on tissue sections of tumor regions with more than 40% viability. We agree that there is tumor heterogeneity and based on the limited data available we were not able to address this in the paper. Based on your comment we have included information in the materials and methods.

 Comment 3: The MSI status is also an important factor in determining the choice of therapy as well as patient outcomes- was this status known and evaluated?

Response: This is an important and relevant point. All the patients in this study were MSI stable, as at our hospital MSI high patients are given immunotherapy and if disease is still present resected however there is a lot of necrosis and we cannot evaluate their HGP. We have indicated in the materials and methods that our patients were MSI stable.

Comment 4: Given that the samples were obtained from the biobank, the mutation analysis was outsourced and as I understand only the histopathology was done for this study in-house by one of the authors, perhaps a more detailed contribution of other authors to the study would be appreciated.

Response: We appreciate this comment and the opportunity to acknowledge the contribution of all the authors. We have indicated in the paper the author contributions.

Reviewer 2 Report

Comments and Suggestions for Authors

This is an interesting manuscript from a study conducted at the McGill University Hospital in Montreal. 

Concerns:

1.  How many patients were excluded because they had previously had liver biopsy?  Were they part of the 352 recruited for the study? If they were part of the 352 enrolled, where in your diagram in figure 1 did you exclude them? What were your list of exclusion criteria besides liver biopsy?

 2. What was your rationale for settling on 352 patients for this study? Was there an opportunity to recruit more than 352 patients for the study? If not, what were the limitations and difficulties?

3. Were all the participants recruited for this study from Montreal area in the province of Quebec or some of them were referred from other regional hospital outside McGill university hospital?

4. What were the racial/ethnic distribution of your study precipitants and does the racial/ethnic profile represent the population of Montreal area?

5. You mentioned few limitations of your study. Do you think those limitations affected the overall conclusion from your study regarding the outcomes of Colorectal cancer liver metastasis (CRCLM) following liver resection?

6. In table 2, were you surprised to see that the major significance differences between desmoplastic and non-desmoplastic  were (1) development of extrahepatic metastasis, (2) development of pulmonary metastasis, (3) no liver metastatic recurrence?

6. In your mutational analysis, you noted that right-sides colon tumors had a statistically  higher incidence of liver tumors with KRAS mutations as compared to the left-sides tumors and rectal tumors.  Please, offer potential explanations.

Comments on the Quality of English Language

Overall, the quality of English language used in the manuscript is excellent and only minor editing may be required.

Author Response

Comment 1: How many patients were excluded because they had previously had liver biopsy?  Were they part of the 352 recruited for the study? If they were part of the 352 enrolled, where in your diagram in figure 1 did you exclude them? What were your list of exclusion criteria besides liver biopsy?

Response: Thank you for pointing this out. To clarify, 66 patients were excluded from the study due to only undergoing intra-operative liver biopsy and not the planned liver resection. In addition a biopsy does not provide sufficient material (ie tumor/liver interface) for an accurate assessment of the HGP. An additional exclusion criterion was any patient under age 18. The liver disease biobank is only available for adult patients over the age of 18. Therefore, no patients were excluded due to the age requirement. These questions are now answered in the first paragraph of “Methods- Patient Selection”.

Comment 2: What was your rationale for settling on 352 patients for this study? Was there an opportunity to recruit more than 352 patients for the study? If not, what were the limitations and difficulties?

Response: At the time of the study, we had data on 352 patients with colorectal cancer liver metastasis who had been recruited into the McGill University Liver Disease biobank from January 2009 through December 2020. There was a small number of additional patients with colorectal liver metastasis seen at McGill University during that time, who did not consent to join the liver disease biobank. Since those patients did not consent to join the liver disease biobank, we were unable to perform histopathologic growth pattern typing on their tumor specimens and thus ineligible for the study. We agree that this was a limitation to our study.

Comment 3: Were all the participants recruited for this study from Montreal area in the province of Quebec or some of them were referred from other regional hospital outside McGill university hospital?

Response: Thank you for your comment as we missed including this in our description. All of the study participants were from the province of Quebec. Many patients were referred to McGill University from smaller hospitals across the province. These questions are now answered in the first paragraph of “Methods- Patient Selection”.

Comment 4: What were the racial/ethnic distribution of your study precipitants and does the racial/ethnic profile represent the population of Montreal area?

Response: Unfortunately, we did not consistently have data points for the race/ethnic distribution of our study population. As you indicated, this is a limitation of the study and has been added to the discussion section.

Comment 5: You mentioned few limitations of your study. Do you think those limitations affected the overall conclusion from your study regarding the outcomes of Colorectal cancer liver metastasis (CRCLM) following liver resection?

Response: Thank you for this comment, however we don’t believe that the limitations affected the overall conclusion. We think this is a strong study that would have been made better if the listed limitations could have been mitigated. This comment has been added to the end of the discussion section.

Comment 6: In table 2, were you surprised to see that the major significance differences between desmoplastic and non-desmoplastic  were (1) development of extrahepatic metastasis, (2) development of pulmonary metastasis, (3) no liver metastatic recurrence?

Response: You raise a very interesting and valid point. As we have been studying HGP for many years we were observing these trends and therefore not surprised when we observed significant differences. These findings complement an earlier study by Neirop et al who previously reported that patients with desmoplastic tumors are more likely to develop liver recurrence amenable to reoperation. We think this is a large part of the reason for why patient survival is improved for patients with desmoplastic tumors. Since desmoplastic tumors are more likely to recur in the liver, patients are more likely to have repeat liver operations or procedures - such as RFA, to control the recurrence. That is in contrast to patients with non-desmoplastic tumors who are more likely to have extra-hepatic disease recurrence which is non-resectable. These findings suggest that overall survival in patients with CRCLM is driven more by rHGP and vessel co-option than dHGP and angiogenesis.  To address your question we have included a comment to the discussion section.

Comment 7: In your mutational analysis, you noted that right-sides colon tumors had a statistically  higher incidence of liver tumors with KRAS mutations as compared to the left-sides tumors and rectal tumors.  Please, offer potential explanations.

Response: You raise a very relevant point in the field. It has been documented in numerous studies that right-sided colon tumors are associated with more advanced tumor stages, increased tumor sizes, and poor differentiation. All these findings are likely due to right-sided tumors becoming symptomatic later than left-sided tumors due to the wider lumen of the left colon and possibly due to the higher difficulty in identification of the right-sided tumors as they are far from the anal verge and unable to be identified on physical exam or sigmoidoscopy. Taking these factors together, right-sided colon tumors have been found in other studies to convey a worse overall survival compared to left-sided colon tumors.

Other studies including Tong et al demonstrated a higher incidence of KRAS mutations in right-sided tumors compared to left-sided colon tumors. Additionally, Brule et all found more mutant KRAS mutations in right-sided colon tumors compared to left-sided colon tumors. However, Zhu et al and the Cancer Genome Atlas Network did not demonstrate any significant differences in right and left colon tumors after performing gene expression profiling. All these studies conclude that further research is needed to identify why right-sided colon tumors are more likely to have KRAS mutations compared to left-sided tumors. To date the mechanistic link between KRAS mutation and tumor sidedness also remains to be assessed. The only significant evidence suggests shorter OS, PFS and DFS in patients with KRAS mutations than in patients with wild type KRAS.

We have made note of this in the discussion.

Round 2

Reviewer 2 Report

Comments and Suggestions for Authors

Thank you for carefully and positively responding to the reviewers' concerns